

# Assessing the impact of two conventional wastewater treatment plants on small streams with effect-based methods

Catalina Trejos Delgado[1], Andrea Dombrowski[1] and Jörg Oehlmann[1,2]

[1] Department Aquatic Ecotoxicology, Johann Wolfgang Goethe Universität Frankfurt am Main, Frankfurt, Germany
[2] Kompetenzzentrum Wasser Hessen, Frankfurt, Germany

Corresponding author
Catalina Trejos Delgado,
mctrejosd@stud.uni-frankfurt.de

## ABSTRACT

Sixty percent of discrete surface water bodies in Europe do not meet the requirements for good ecological and chemical status and in Germany, the situation is even worse with over 90% of surface water bodies failing to meet the threshold. In addition to hydromorphological degradation, intensive land use and invasive species, chemical pollution is primarily considered to be responsible for the inadequate ecological status of the water bodies. As a quantitatively important source of micropollutants, wastewater treatment plants (WWTPs) represent an important entry path for chemical stressors. It is therefore important to analyze the effectiveness of the WWTPs in eliminating micropollutants and other chemical stressors to mitigate the negative impacts of the treated wastewater (WW) in aquatic ecosystems. Accordingly, in this study, we evaluated the impacts of two conventional, medium-sized WWTPs on their small receiving water systems in the southwestern region of Hessen in Germany during two sampling campaigns (spring and fall) using effect-based methods (EBM). We hypothesized that due to the insufficient elimination of micropollutants, a broad spectrum of toxic effects would be detected in conventionally treated WW and also in the receiving surface waters downstream the WWTPs. As EBMs a battery of *in vitro* assays and active biomonitoring using two *in vivo* assays were applied. The results supported our hypothesis and showed that the untreated WW had a very high baseline toxicity and also high endocrine and mutagenic activities. Conventional WW treatment, consisting of mechanical and biological treatment with nitrification, denitrification and phosphate precipitation, reduced baseline toxicity by more than 90% and endocrine activities by more than 80% in both WWTPs. Despite these high elimination rates, the remaining baseline toxicity, the endocrine, dioxin-like and mutagenic activities of the conventionally treated WW were so high that negative effects on the two receiving waters were to be expected. This was confirmed in the active monitoring with the amphipod *Gammarus fossarum* and the mudsnail *Potamopyrgus antipodarum*, as mortality of both species increased downstream of the WWTPs and reproduction in *P. antipodarum* was also affected. These results indicate that advanced WW treatment is needed to more effectively eliminate chemical stressors to prevent negative impacts of treated WW particularly in small receiving waters.

## INTRODUCTION

In many areas worldwide, the ecological status of small streams is impacted by morphological degradation (*e.g.*, channelization and straightening), agricultural land use in the catchment, and a high load of treated wastewater (WW) as the major source of aquatic pollution (*Reemtsma et al., 2006*; *Stalter et al., 2013a*; *Stalter et al., 2013b*; *Ternes et al., 2017*). Next to other anthropogenic chemicals entering the water cycle *via* municipal and industrial wastewater treatment plants (WWTPs), micropollutants, which include pharmaceuticals, personal care products, biocides, pesticides, and endocrine disrupting chemicals (EDCs), among others, have been identified as main drivers that, even at low concentrations, can affect ecosystems and human health (*Khan et al., 2022*; *Yang et al., 2022*). These stressors have led to significant declines in aquatic biodiversity, which in turn has had a profound impact on the ecological integrity of many aquatic ecosystems (*Enns et al., 2023*; *Jähnig et al., 2021*). Currently, 60% of the discrete surface water bodies in Europe do not meet the requirements of a good ecological and chemical status according to the European Water Framework Directive (*European Commission, 2019*). In Germany the situation is even worse because more than 90% of surface waters have not yet reached a good ecological status (*German Environment Agency, 2017*).

Municipal WWTPs were designed to reduce high nitrogen, phosphorus, and organic matter levels. However, many studies have shown that the elimination rate for micropollutants often remains poor (*Giebner et al., 2018*; *Gosset et al., 2021*; *Long & Bonefeld-Jørgensen, 2012*). These pollutants are thus continuously released in trace amounts (typically from ng/L to µg/L) into receiving watercourses (*Wolf et al., 2022*). Particularly, estrone, 17β-estradiol, 17α-ethinylestradiol and bisphenol A have been identified by various studies both in the influent and in the effluent of WWTPs (*Körner et al., 2000*; *Kusk et al., 2011*). Besides other compounds with mutagenic potential, polycyclic aromatic hydrocarbons (PAHs), have been measured in industrial and domestic discharges (*Denison & Nagy, 2003*; *Reifferscheid et al., 2011*). Additionally, biologically highly active chemicals have been detected in the effluents from WWTPs on a regular basis (*Lopez et al., 2022*; *Ternes, Joss & Oehlmann, 2015*; *Ternes et al., 2017*).

Due to the prominent role of WWTPs as a source of micropollutants in surface waters, there are currently significant efforts to improve the elimination of micropollutants in WW treatment. Oxidative processes such as ozonation and filtration processes with activated carbon have proven to be particularly promising (*Prasse et al., 2015*). Also in our study area, the Hessian Ried south of Frankfurt am Main, Germany, WW-borne micropollutants were detected not only in high concentrations in the receiving surface waters, but also in the groundwater (*Hessian Ministry for the Environment, Climate Protection, Agriculture and Consumer Protection (HMUKLV), 2018b*). Concentrations of up to 1.9 µg/L for diclofenac, 1.2 µg/L for carbamazepine and 1.4 µg/L for benzophenone-4 have been found in the surface waters. With the "Hessian Ried Trace Substance Strategy", the Hessian state government is currently implementing various measures to reduce micropollutant concentrations in surface waters and groundwater and thereby minimize the ecological consequences and risks to drinking water resources. A central measure is the upgrading

of selected municipal WWTPs with an advanced WW treatment step (so-called 4th treatment step) to achieve improved micropollutant elimination (*Hessian Ministry for the Environment, Climate Protection, Agriculture and Consumer Protection (HMUKLV), 2018a*).

As part of a scientific accompanying program, we will examine the effectiveness of this measure, which will be implemented in the next years, with effect-based methods (EBMs). EBMs are particularly suitable for monitoring programs because they integrate the effects of chemical mixtures or chemicals that are not analyzed, are able to identify specific drivers of toxicity and link the exposure levels to chemicals directly to biological effects. They are therefore good screening tools to assess the relevance of the pressures and impacts on water bodies, to establish early warning systems and to provide additional support in water and sediment quality assessment in supplement to conventional chemical and ecological monitoring (*Wernersson et al., 2015*). For that reason, the aim of our whole project is to describe the situation before and after the implementation of the 4th treatment step with a combination of ozonation, followed by activated carbon filtration in two WWTPs. In this first step, we establish a baseline (current pre-upgrade conditions) to characterize both the impact profile of the treated WW and the effects of WW discharge on the receiving surface waters. In this way, the current status quo is analyzed, which makes it possible to record an improvement in the contamination and effect level in future investigations after the implementation of the WWTP upgrade. We hypothesize that due to the insufficient elimination of micropollutants (1) a broad spectrum of baseline toxic and specific toxic activities can be detected in conventionally treated WW using EBMs and (2) the corresponding effects can also be detected in the receiving surface water downstream the WWTP. As EBMs we use a battery of *in vivo* and *in vitro* bioassays: the baseline or unspecific toxicity is assessed with (1) the Microtox assay with *Aliivibrio fischeri*, (2) mutagenicity with the Ames fluctuation assay (*Salmonella typhimurium*), (3) endocrine and dioxin-like activities with recombinant yeast reporter gene assays *(Saccharomyces cerevisiae)*, and (5) reproduction and survival in an active biological effect monitoring with the stream amphipod *Gammarus fossarum* and (6) the mudsnail *Potamopyrgus antipodarum* in two sampling campaigns (spring and fall) to take into account possible seasonal effects.

## MATERIALS & METHODS

### Characterization of the wastewater treatment plants Mörfelden-Walldorf and Bickenbach

The medium-sized WWTPs Mörfelden-Walldorf (MW) and Bickenbach (B) are located in the southwestern region of Hessen in Germany and have approximately 48,000 and 32,000 population equivalents (PE), respectively. The WW is currently processed with conventional treatment (mechanical, biological treatment, nitrification, denitrification, phosphate precipitation) in both WWTPs. The receiving watercourses for the treated WW are the Geräthsbach (waterbody code 239818) for the WWTP MW, and the Landbach (waterbody code 239628) for the WWTP B. They are classified as small streams in riverine floodplains (type 19) and exhibit poor chemical status, strongly or

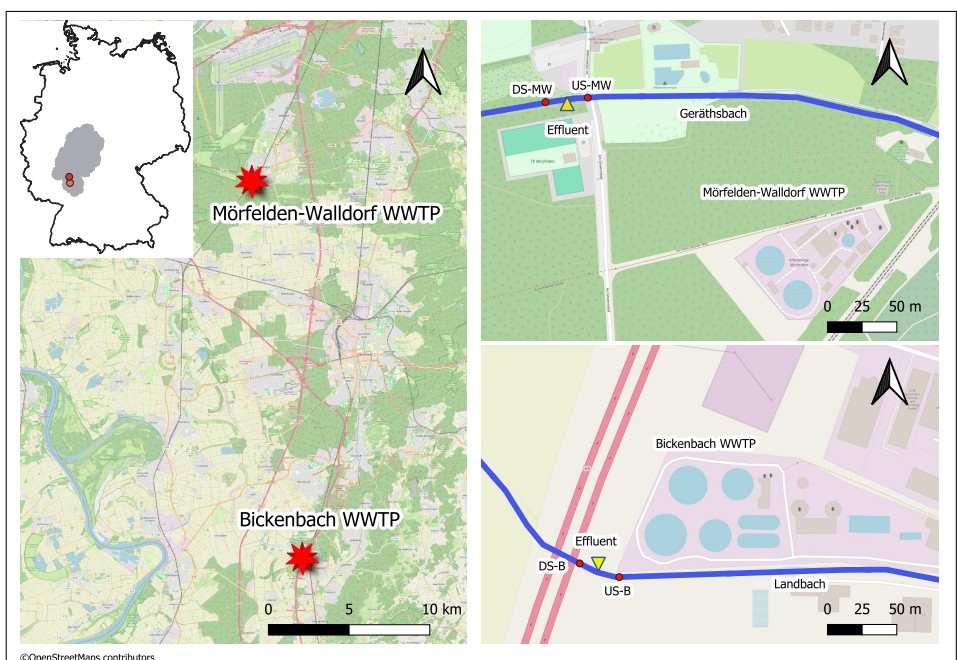

**Figure 1** **Map of the sampling location of the two WWTPs in Germany (exact positions are provided in Table 1).** This figure was created with map data from OpenStreetMap (http://openstreetmap.org) and can be used and modified under CC-by-SA-License (http://creativecommons.org).

completely altered morphological structure, as well as moderate and poor ecological status, respectively (*Hessisches Landesamt für Naturschutz, Umwelt und Geologie (HLNUC), 2022*). Additionally, the mean flow (MQ) in these streams is 238 L/s (Geräthsbach) and 125 L/s (Landbach) (*Hessisches Landesamt für Naturschutz, Umwelt und Geologie (HLNUC), 2022*). Ten km upstream WWTP MW, there is another WWTP in Langen (Fig. 1).

## Collection and storage of samples

For the study, four sampling sites were chosen for each of the two WWTPs (Table 1).

In two sampling campaigns in April/May 2022 (spring) and October/November 2022 (fall), 2.1 L surface water samples were taken upstream (US) and downstream (DS) the WWTPs in the receiving rivers, and WW samples were taken after the mechanical (MT) and biological treatment (BT) at each WWTP at the beginning and at the end of the four weeks active biomonitoring period. Accordingly, two water samples per site were obtained and analyzed per season. During the sampling campaign in fall 2022, the outflow of the WWTP in Langen, which is located 10 km upstream of WWTP MW at the Geräthsbach, was also considered (sample code BT-L), because the first collection campaign had shown that the Geräthsbach already showed high activities in the EBM upstream the WWTP MW. At each sampling site, physicochemical parameters including temperature, pH, dissolved oxygen, and conductivity were measured in the field with a portable multi-meter (HQ40d, Hach, Germany). The water samples were collected in glass vessels precleaned with deionized water, acetone, and ethanol and heated to 200 °C before use. The samples were transported
**Table 1 Description of sampling sites.** Sampling sites at the WWTPs Mörfelden-Walldorf and Bickenbach and in the receiving rivers.

| Sampling site [code] | Characteristic |
|---|---|
| Upstream [US-MW] | Geräthsbach 20 m upstream of the effluent of WWTP Mörfelden-Walldorf (49°58′14.6″N 8°32′59.3″E) |
| Mechanical treatment [MT-MW] | 24 h composite sample after mechanical treatment at WWTP Mörfelden-Walldorf |
| Biological treatment [BT-MW] | 24 h composite sample after biological treatment and phosphate precipitation at WWTP Mörfelden-Walldorf |
| Downstream [DS-MW] | Geräthsbach 20 m downstream of the effluent of WWTP Mörfelden-Walldorf (49°58′14.7″N 8°32′56.6″E) |
| Upstream [US-B] | Landbach 20 m upstream of the effluent of WWTP Bickenbach (49°45′33.9″N 8°35′52.0″E) |
| Mechanical treatment [MT-B] | 72 h composite sample after mechanical treatment at WWTP Bickenbach |
| Biological treatment [BT-B] | 72 h composite sample after biological treatment and phosphate precipitation at WWTP Bickenbach |
| Downstream [DS-B] | Landbach 20 m downstream of the effluent of WWTP Bickenbach (49°45′33.8″N 8°35′49.4″E |

and stored at 7 °C in darkness. Phosphate, ammonium, nitrite, and nitrate concentrations were determined using spectroquant test kits (Merck, Darmstadt, Germany) immediately after bringing the samples to the laboratory.

## Pretreatment of samples

Water samples were filtered through glass microfiber filters (VWR International GmbH, No. 692, European Cat. No. 516-0885, 90 mm, particle retention: 1.5 μm, Darmstadt, Germany). Then, 2 L of each water sample was solid-phase extracted (SPE) with OASIS HLB cartridges (6cc, 200 mg; Waters, Milford, MA, USA) according to *Giebner et al. (2018)* and transferred to a final volume of 400 μL DMSO so that the final extracts were 5,000-fold concentrated compared to the aqueous samples. Extracts were stored in glass vials at −25 °C until further processing. To check a potential contamination of the samples during the sample preparation, 2 L of ultrapure water was extracted in parallel (SPE blank) (*Giebner et al., 2018*).

## Analysis of samples with effect-based methods
### *In vitro assays*

*Baseline toxicity test.* The Microtox assay or bioluminescence inhibition test with the bacterium *Aliivibrio fischeri* was conducted to assess the baseline toxicity in the extracts. Based on the International Organization for Standardization (ISO) guideline 11348-3 (*ISO, 2007*) the assay was performed in 96-well microplates according to *Völker et al. (2017)*. Samples were analyzed in dilution series in a saline buffer with seven consecutive steps (spacing factor 2 between concentrations), resulting in a 25-fold to 0.2-fold final concentration (*Völker et al., 2017*). Negative and solvent controls, SPE blank, reference compound (3,5-dichlorophenol), and SPE extracts were serially diluted (1:2) in a saline buffer. 100 μL sample was added to 50 μL of *A. fischeri* solution (not exceeding 1% DMSO

in the final medium volume). To detect inhibition, luminescence was measured prior to sample addition and after 30 min incubation using a microplate reader (Infinite 200 Pro; Tecan, Crailsheim, Germany) as described by *Völker et al. (2017)*. The inhibition of luminescence is expressed as the arithmetic mean value of the 50% effect concentration ($EC_{50}$) of three independent experiments, referring to the relative enrichment factor (REF) of the water sample. Nontoxic water samples exhibit an $EC_{50}$ threshold value of 300 REF.

*Recombinant yeast reporter gene assays.* The Yeast Estrogen Screen (YES), Yeast Androgen Screen (YAS), Yeast Anti-Estrogen Screen (YAES), Yeast Anti-Androgen Screen (YAAS), and Yeast Dioxin Screen (YDS) were used to assess the receptor-mediated agonistic and antagonistic endocrine activity of the extracted water samples as well as the activation of the aryl hydrocarbon receptor assay according to *Brettschneider et al. (2019a)*; *Brettschneider et al. (2019b)*, *Giebner et al. (2018)* and *Schneider et al. (2020)*. These assays use genetically modified strains of *Saccharomyces cerevisiae*, which contain the human estrogen receptor α, androgen and aryl hydrocarbon receptor, respectively. The YAES and YAAS were performed with native samples (filtrated surface water or WW samples without SPE processing) because SPE enrichment of water samples from surface waters and WWTPs leads to a strong loss of anti-estrogenic and anti-androgenic activities (*Brettschneider et al., 2019a*; *Giebner et al., 2018*; *Schneider et al., 2020*). The YES, YAS and YDS were performed in three independent experiments with every sample, the YAES and YAAS in two independent experiments, with eight technical replicates per sample. The measured activities are expressed as equivalent concentrations (EQ) for the positive substances 17β-estradiol (YES), testosterone (YAS), 4-hydroxytamoxifen (YAES), flutamide (YAAS) and β-naphthoflavone (YDS) and were corrected regarding dilution and enrichment so that equivalent concentrations relate back to the native water sample.

*Ames fluctuation test.* The Ames fluctuation assay (*ISO, 2012*) was performed to assess the mutagenic potential of the SPE samples, using *Salmonella typhimurium* strains YG1041 and YG1042 (*Hagiwara et al., 1993*) with and without metabolic activation by rodent liver enzymes (S9-mix; Envigo CRS, Roßdorf, Germany).

The test is based on the inability of genetically modified *S. typhimurium* strains to synthesize the essential amino acid histidine and, therefore, not being able to survive in a histidine-free medium (*Maron & Ames, 1983*). The bacteria are exposed to the SPE extracts in a 16.7-fold final sample concentration (0.2% v/v solvent) and incubated for 72 h at 37 °C in a histidine-free medium. After this time, a color change of the medium indicates the survival of the bacteria by regaining the ability to synthesize histidine because of a back mutation.

YG1041 and YG1042 strains without the S9 mix were tested with 2-nitrofluorene as a positive control. Conversely, these strains with S9-mix were tested with 2-amino anthracene as a positive control. The mutagenic potential of the samples was assessed *via* the percentage of reverted wells, subtracted by the mutants of negative historical controls. Samples exceeding the threshold of 20.8% revertant wells were considered as mutagenic.

Every sample was tested with both strains, with and without the S9-mix. If an effect was observed, the test was consequently conducted two times.

### In vivo assays

The *in vivo* assays were applied as an active *in-situ* monitoring at the upstream and downstream samplings sites of both WWTPs as described by *Brettschneider et al. (2019a)* and *Harth et al. (2018)*.

*Active monitoring with Gammarus fossarum.* The gammarids for the active monitoring were collected at the upper reach of the river Urselbach (Hessen, Germany, 50°13′30.1″N 8°31′06.1″E) 5 days before the start of the spring and fall campaigns. This site was selected because the upper reach of the Urselbach is entirely natural and free of anthropogenic inputs. Also, a genetic characterization of the *G. fossarum* population has shown that only *G. fossarum* type B (or clade 11; *Weigand et al., 2020*) occurs there. After sampling, gammarids were transported to the laboratory and kept in Urselbach stream water in a temperature-controlled room at 12 °C with oxygen supply until the start of the active monitoring campaign. At the US and DS sampling sites of both WWTPs, two stainless steel cages each containing three enclosures (12.5 cm × 6.0 cm) were fixed on the riverbed in the direction of flow. The enclosures were capped with a net (mesh size 1.0 mm) at the ends to ensure a water flow. Each enclosure, representing a replicate, contained ten gammarids, resulting in a total of 60 gammarids per site which were exposed *in situ* for four weeks. Black alder leaves (*Alnus glutinosa*) were provided as food in the enclosures *ad libitum* and two small nets (polytetrafluorethylene, 8.2 cm × 3.3 cm) served as hiding places to prevent cannibalism.

After four weeks of exposure, the cages were taken to the laboratory, and the gammarids were fixed in 70% ethanol and stored separately in Eppendorf tubes. As biological endpoints the mortality and fecundity index (number of eggs divided by the body length) were assessed.

For all measurements, a stereomicroscope (Olympus SZ61 R, Olympus Corporation, Tokio, Japan) with a digital camera (JVC Digital Camera KY-F75U, Victor Company of Japan Ldt., Yokohama, Japan) and the Software Diskus (Version 4.50.1458, Carl H. Hilgers, Königswinter, Germany) was used.

*Active monitoring with Potamopyrgus antipodarum.* The mudsnails were collected at the rivers Lumda and Horloff (Hessen, Germany) in spring and at the rivers Gambach and Waschbach (Hessen, Germany) in fall. These sites were selected because the upper reaches of these streams are free of anthropogenic inputs. Snails with a shell length between 3.5 and 4.5 mm were used, following the protocol by *Brettschneider et al. (2019a)*. At the US and DS sampling sites of both WWTPs, two stainless steel cages containing three smaller steel cages (4.5 cm × 3.5 cm, mesh size 0.7 mm) were fixed on the riverbed in the direction of flow. Ten snails were placed in each small cage, representing the six replicates and resulting in a total of 60 snails which were exposed *in situ* for four weeks. Organic carrot cubes served as food supply and were refilled after two weeks.

As biological endpoints the mortality of the snails, the shell length and the number of embryos of the surviving snails were determined using a stereomicroscope with an ocular micrometer (SMZ-168; Motic, Xiamen, China). The active monitoring with *P. antipodarum* followed the principle of the OECD guideline 242 (*OECD, 2016*) to assess the impact of reproductive toxicants, including EDCs, on this species.

## Statistical analyses

Statistical analyses were performed using GraphPad Prism version 5.03 for Windows (GraphPad Software, San Diego, California, USA). Normal distribution was tested using the D'Agostino and Pearson test. Continuous data were analyzed with t-tests for normally distributed data and Welch's correction for unequal variances. All data were tested as unpaired data and with 95% confidence interval. Quantal data (mortality) were analyzed with Fisher's exact test. The level of significance was defined as $\alpha = 0.05$ and indicated in the graphs with asterisks as follows: $* = p < 0.05$; $** = p < 0.01$ and $*** = p < 0.001$.

# RESULTS

## Physicochemical parameters

The conventional treatment process (BT *vs.* MT) in both WWTPs removed more than 90% of the ammonium and phosphate and around 33% of the conductivity, independent of the season. Moreover, the oxygen saturation increased significantly during the fall campaign at the WWTP Mörfelden-Walldorf (MW) by 79% and in spring at the WWTP Bickenbach (B) by 89%. In the BT sample the ammonium concentrations were significantly reduced compared to MT, but the nitrate concentrations increased significantly by factor 9.6 (spring) to 19.6 (fall) in MW and by factor 2.5 (spring) to 2.9 times (fall) in B. Considering the sum concentrations of ammonium-N, nitrite-N and nitrate-N, inorganic N was removed by 90% and 94% in MW and 97 and 94% in B in spring and fall, respectively.

### *In vitro* assays
### *Baseline toxicity test*

The baseline toxicity, as determined with the Microtox assay, was significantly higher DS compared to US, except at WWTP MW in fall (Fig. 2). Furthermore, all sample sites, except US-B in both seasons showed a significant baseline toxicity. The baseline toxicity was removed by 90% in the WWTPs MW and B in spring and by 96% and 97% in fall. After including the effluent from the WWTP Langen (BT-L in Fig. 2A; located upstream of the WWTP MW) in the analyses in fall, it became apparent that the baseline toxicity, which was already significantly increased at US-MW, can be attributed to the discharges from the WWTP Langen.

### *Estrogenic and anti-estrogenic activity*

All samples exhibited a significant estrogenic activity above the limit of quantification (LOD = 0.148 ng E-EQ/L). The estrogenic activity increased significantly at the DS sites compared to US at both WWTPs, with the exception of MW WWTP in fall which showed a 6% decrease (Fig. 3). Despite over 90% removal by WWTPs, estrogenic activity was 22%

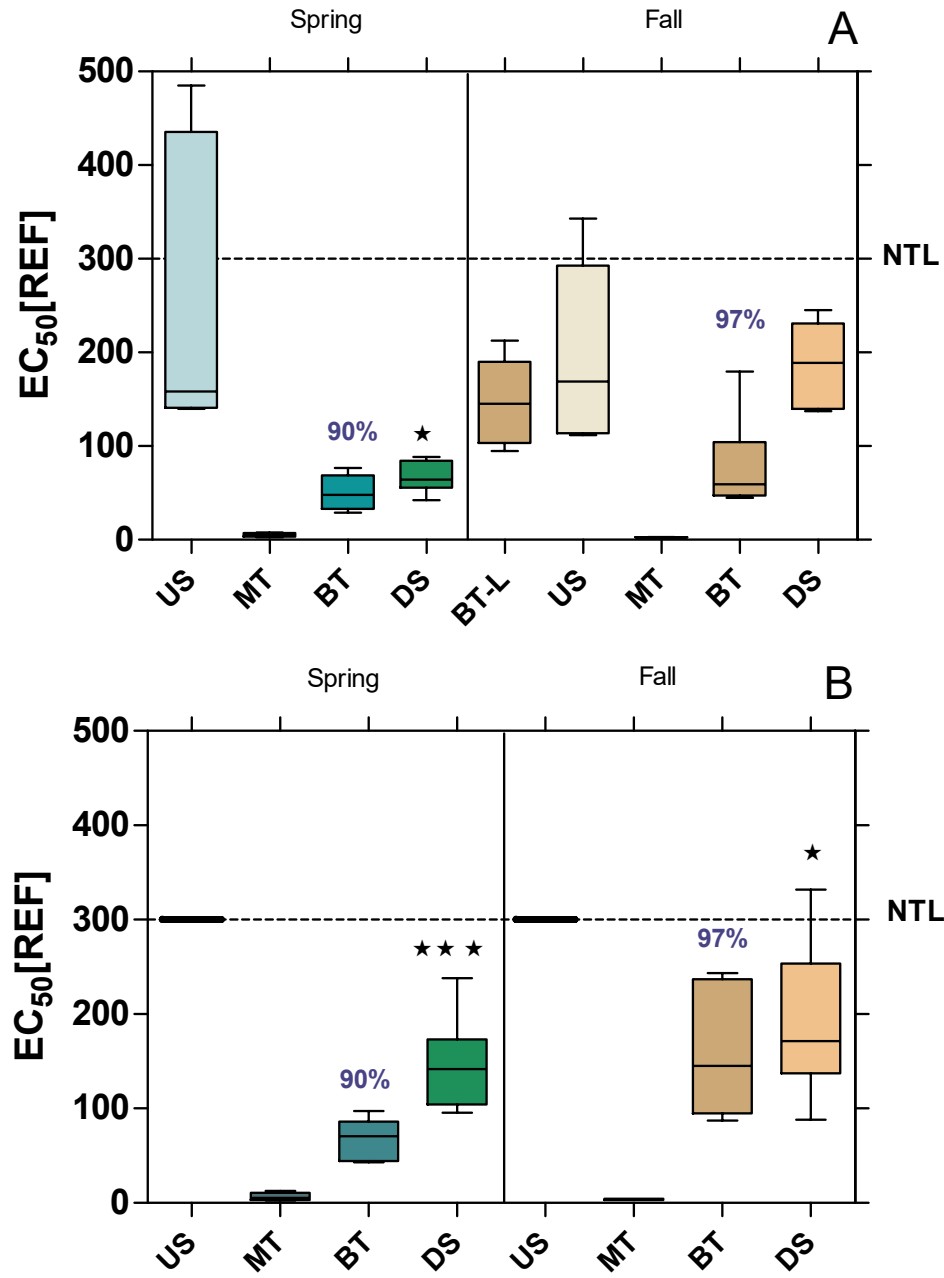

**Figure 2** **Baseline toxicity of water samples at the WWTPs Mörfelden-Walldorf (A) and Bickenbach (B) and in the receiving streams.** Toxicity was determined by the Microtox assay and is expressed as EC50 values, referring to the relative enrichment factor (REF) of the water. Low values indicate higher toxicity. Dashed line: non toxic limit (NTL). Significant differences refer to downstream values (DS) compared to upstream values (US) (unpaired $t$-test with Welch's correction for unequal variances; ★ $p < 0.05$, ★★ $p < 0.01$, ★★★ $p < 0.001$). Numbers in blue indicate the mean decrease in % of measured baseline toxicity during the biological treatment.

higher at the WWTP MW in spring, and 60% at the WWTP B in both seasons compared to their respective US sites.

The WWTPs did not cause significant increase in anti-estrogenic activity in the receiving streams (Fig. 4). Both WWTPs had a high removal efficiency of anti-estrogenic activities of over 80% in MW and 90% in Bickenbach in both seasons.

### Androgenic and anti-androgenic activity

Androgenic and anti-androgenic activities were analyzed in 5,000-fold concentrated SPE extracts and native water samples, respectively. While no cytotoxic effects occurred in the native water samples, the observed cytotoxicity in the SPE extracts necessitated the analysis of diluted extracts. With the exception of the MT samples, no other sample from the two WWTPs and in the receiving streams showed an androgenic and anti-androgenic activity with a LOD of 25.6 ng testosterone-EQ/L and 771 μg flutamide-EQ/L, respectively during the spring and fall campaigns.

### Dioxin-like activity

The MT samples were also cytotoxic for the yeast cells in the YDS, so that the dioxin-like activity had to be assessed in serial dilutions (Fig. 5). However, in the case of Bickenbach, even with a dilution series, it was not possible to calculate the dioxin-like activity of the MT samples (Fig. 5B).

All sample sites showed dioxin-like activity above the limit of detection (LOD = 33.8 ng β-naphthoflavone-EQ/L). Dioxin-like activity increased significantly in the DS compared to the US samples, except for the fall campaign at the WWTP MW (Fig. 5A). The increases were 21% and 11% in MW and 85% and 72% in B during the spring and fall campaigns, respectively. The calculation of the removal efficiency of dioxin-like activity was only possible for the WWTP MW because of the cytotoxicity even with highly diluted MT samples of the WWTP B. In MW, elimination rates of 60% and 53% were observed.

### Ames fluctuation test

The tests with both strains without the S9-mix resulted in negative results for all samples (less than 20.8% revertant colonies). Due to their high cytotoxicity, most MT samples from the WWTPs could not be analyzed with the Ames fluctuation test. All analyzed BT samples from the WWTPs MW, B and Langen proved to be mutagenic in the tests with the strains YG1041 and YG1042 with S9 mix, as did the DS samples (Table 2). The US samples of the WWTP MW were also mutagenic with both strains, while the US sample at WWTP B was not mutagenic in the spring campaign, but mutagenic with the YG1042 strain with S9-mix in the fall campaign.

### Active monitoring with Gammarus fossarum

In both rivers the mortality in US was higher in fall than in spring (Fig. 6). There was also a significantly higher mortality in DS ($p < 0.05$) compared to US (except WWTP B in fall) with an increase of 66% and 77% at WWTP MW and B, respectively.

The fecundity index was assessed for the spring campaign but did not show any significant difference between DS and corresponding US. For the fall campaign, the fecundity index

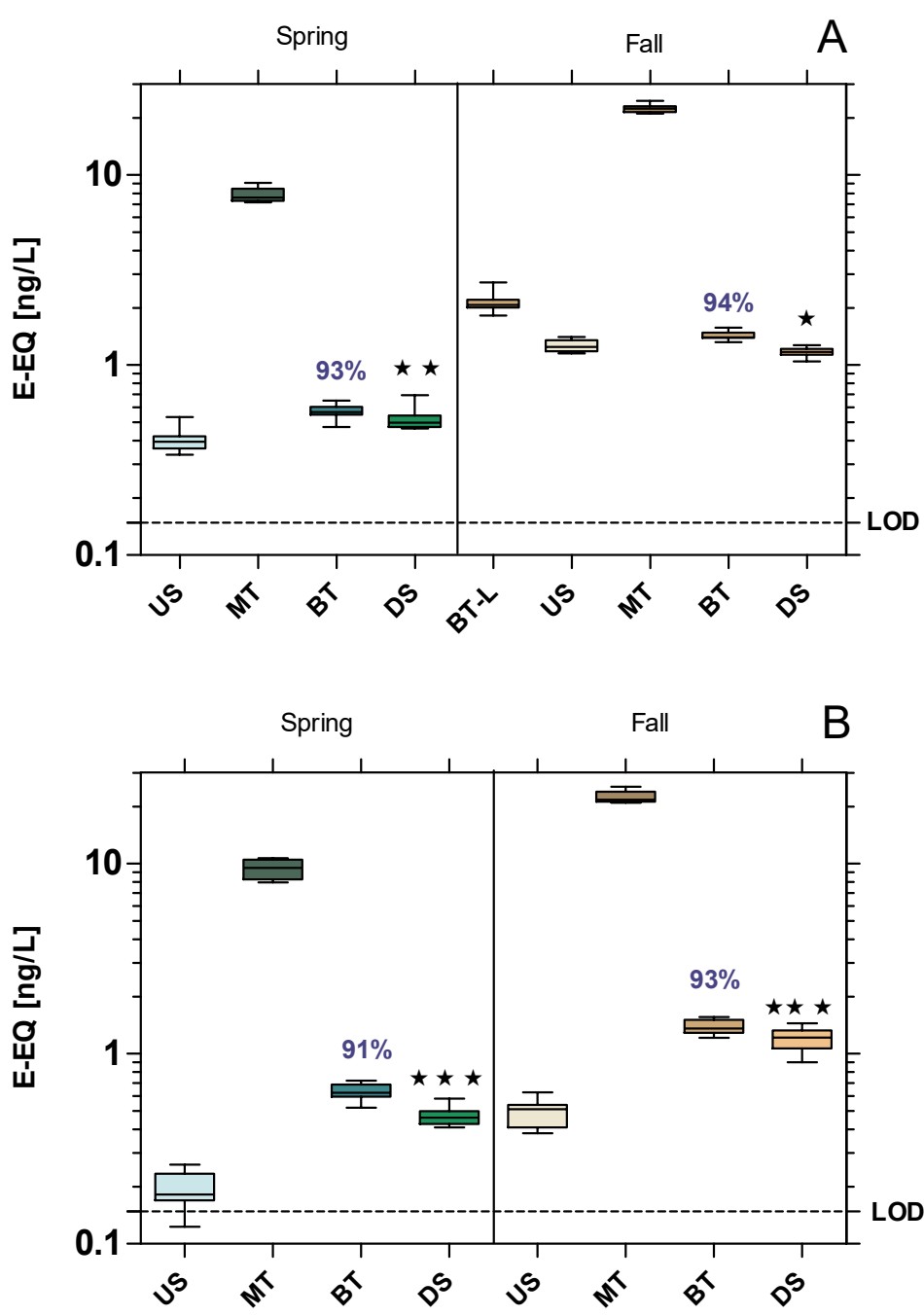

**Figure 3** **Estrogenic activity at the WWTPs Mörfelden-Walldorf (A) and Bickenbach (B) and the receiving rivers, determined with the Yeast Estrogen Screen (YES).** Activities are expressed as 17β-estrogen equivalent concentrations (E-EQ). Dashed line: limit of quantification (LOD: 0.148 ng E-EQ/L). Significant differences refer to downstream values (DS) compared to upstream values (US) (unpaired *t*-test with Welch's correction for unequal variances; ★ $p < 0.05$, ★★ $p < 0.01$, ★★★ $p < 0.001$). Numbers in blue indicate the mean decrease in % of measured estrogenic activity during the biological treatment.

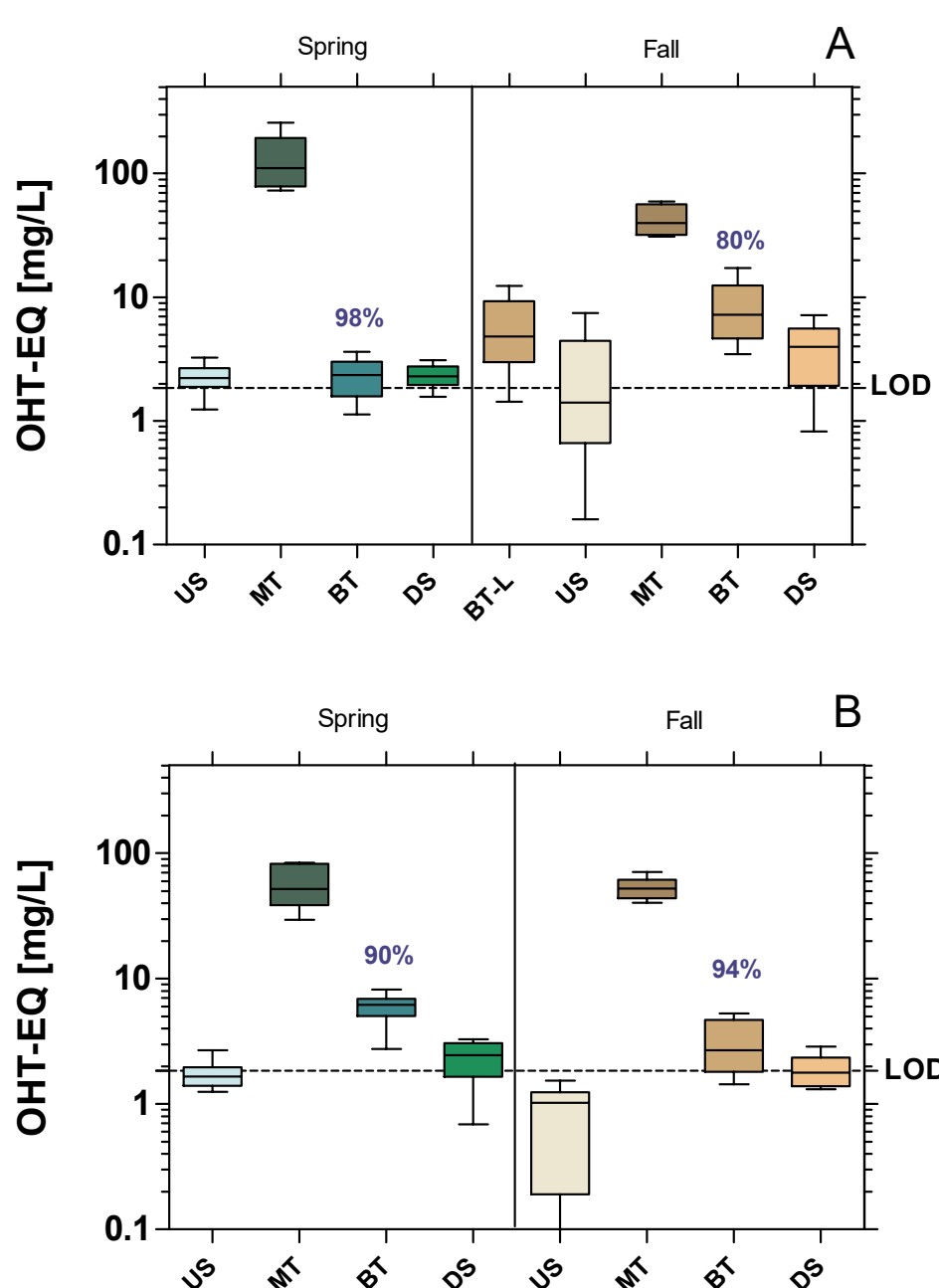

**Figure 4** Anti-estrogenic activity at the WWTPs Mörfelden-Walldorf (A) and Bickenbach (B) and the receiving rivers, determined with the Yeast Anti Estrogen Screen (YAES). Activities are expressed as hydroxy-tamoxifen equivalent concentrations (OHT-EQ). Dashed line: limit of quantification (LOD: 1.84 mg OHT-EQ/L). None of the downstream values (DS) exhibits statistically significant differences to upstream values (US) (unpaired *t*-test with Welch's correction for unequal variances; *p* > 0.05). Numbers in blue indicate the mean decrease in % of measured estrogenic activity during the biological treatment.

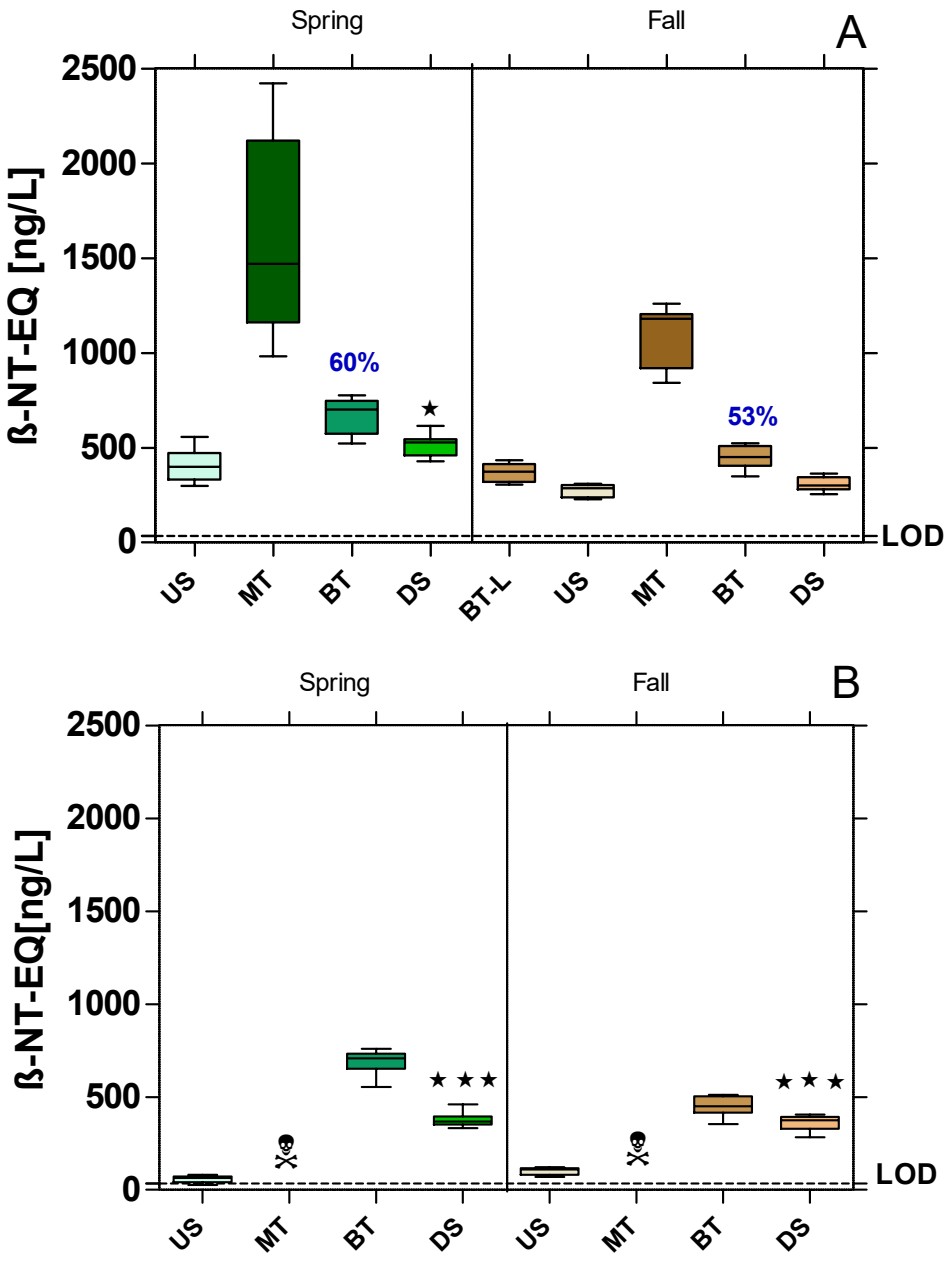

**Figure 5** **AhR agonistic activity at the WWTPs Mörfelden-Walldorf (A) and Bickenbach (B) and in the receiving rivers, determined with the Yeast Dioxin Screen (YDS).** Activities are expressed as β-naphthoflavone equivalent concentrations (β-NF-EQ). Dashed line: limit of quantification (LOD: 33.8 ng β-NF-EQ/L). Significant differences refer to downstream values (DS) compared to upstream values (US) (unpaired *t*-test with Welch's correction for unequal variances; ★ $p < 0.05$, ★★★ $p < 0.001$). Numbers in blue indicate the mean decrease in % of measured estrogenic activity during the biological treatment; skull and crossbones symbol indicate cytotoxic samples.

could not be calculated because the gammarid populations were outside their reproductive phase.

**Table 2 Overview of the results of the mutagenicity test.** Mutagenicity at the WWTPs Mörfelden–Walldorf and Bickenbach in the Ames fluctuation test with *Salmonella typhimurium* strains YG1041 and YG1042 with and without S9 mix. Mutagenic samples (>20.8% revertant colonies) are labelled with (+), non-mutagenic samples with (-). n.a., not analyzed due to high cytotoxicity.

| Season | WWTP | Sample site | Test response | Revertant percentage mean (%) |
|--------|------|-------------|---------------|-------------------------------|
| | | **YG1041-S9 mix** | | |
| | Mörfelden Walldorf | US | – | 12.9 |
| | | MT | – | 10.8 |
| | | BT | – | 14.0 |
| | | DS | – | 16.1 |
| Spring | Bickenbach | US | – | 8.7 |
| | | MT | – | 15.1 |
| | | BT | – | 20.2 |
| | | DS | – | 11.9 |
| | Mörfelden Walldorf | BT-L | – | 10.8 |
| | | US | – | 12.9 |
| | | MT | – | 14.4 |
| | | BT | – | 14.0 |
| Fall | | DS | – | 16.1 |
| | Bickenbach | US | – | 8.8 |
| | | MT | – | 12.2 |
| | | BT | – | 20.2 |
| | | DS | – | 11.9 |
| | | **YG1041+S9 mix** | | |
| | Mörfelden Walldorf | US | + | 39.3 |
| | | MT | n.a. | n.a. |
| | | BT | + | 49.7 |
| | | DS | + | 56.5 |
| Spring | Bickenbach | US | – | 4.8 |
| | | MT | n.a. | n.a. |
| | | BT | + | 49.2 |
| | | DS | + | 52.3 |
| | Mörfelden Walldorf | BT-L | + | 52.8 |
| | | US | + | 22.1 |
| | | MT | n.a. | n.a. |
| | | BT | + | 57.5 |
| Fall | | DS | + | 54.4 |
| | Bickenbach | US | – | 15.3 |
| | | MT | n.a. | n.a. |
| | | BT | + | 47.1 |
| | | DS | + | 53.3 |

**Table 2** (*continued*)

| | | YG1042-S9 mix | | |
|---|---|---|---|---|
| | | US | – | 17.8 |
| | Mörfelden Walldorf | MT | n.a | n.a |
| | | BT | – | 19.1 |
| Spring | | DS | – | 17.8 |
| | | US | – | 10.9 |
| | Bickenbach | MT | – | 5.8 |
| | | BT | – | 17.3 |
| | | DS | – | 13.7 |
| | | BT-L | n.a | n.a |
| | | US | – | 17.8 |
| | Mörfelden Walldorf | MT | n.a | n.a |
| | | BT | – | 9.5 |
| Fall | | DS | – | 17.9 |
| | | US | – | 10.9 |
| | Bickenbach | MT | – | 5.8 |
| | | BT | – | 17.3 |
| | | DS | – | 13.7 |
| | | YG1042+S9 mix | | |
| | | US | + | 41.9 |
| | Mörfelden Walldorf | MT | n.a. | n.a. |
| | | BT | + | 47.1 |
| Spring | | DS | + | 41.4 |
| | | US | – | 20.5 |
| | Bickenbach | MT | n.a. | n.a. |
| | | BT | + | 45.0 |
| | | DS | + | 42.4 |
| | | BT-L | + | 54.9 |
| | | US | + | 32.5 |
| | Mörfelden Walldorf | MT | n.a. | n.a. |
| | | BT | + | 52.8 |
| Fall | | DS | + | 51.3 |
| | | US | + | 36.2 |
| | Bickenbach | MT | n.a. | n.a. |
| | | BT | + | 51.8 |
| | | DS | + | 56.0 |

### Active monitoring with *Potamopyrgus antipodarum*

The mortality increased significantly at DS at both WWTPs in fall, while there was no effect on mortality during the spring campaign (Fig. 7).

The embryo numbers as the parameter for reproduction in *P. antipodarum* increased at DS of both WWTPs in the spring campaign, although this increase was not statistically significant (Fig. 8). In contrast, the embryo numbers at DS of the WWTP MW were on the same level as at US in the fall campaign but increased significantly ($p < 0.01$) at the WWTP B in the same period.

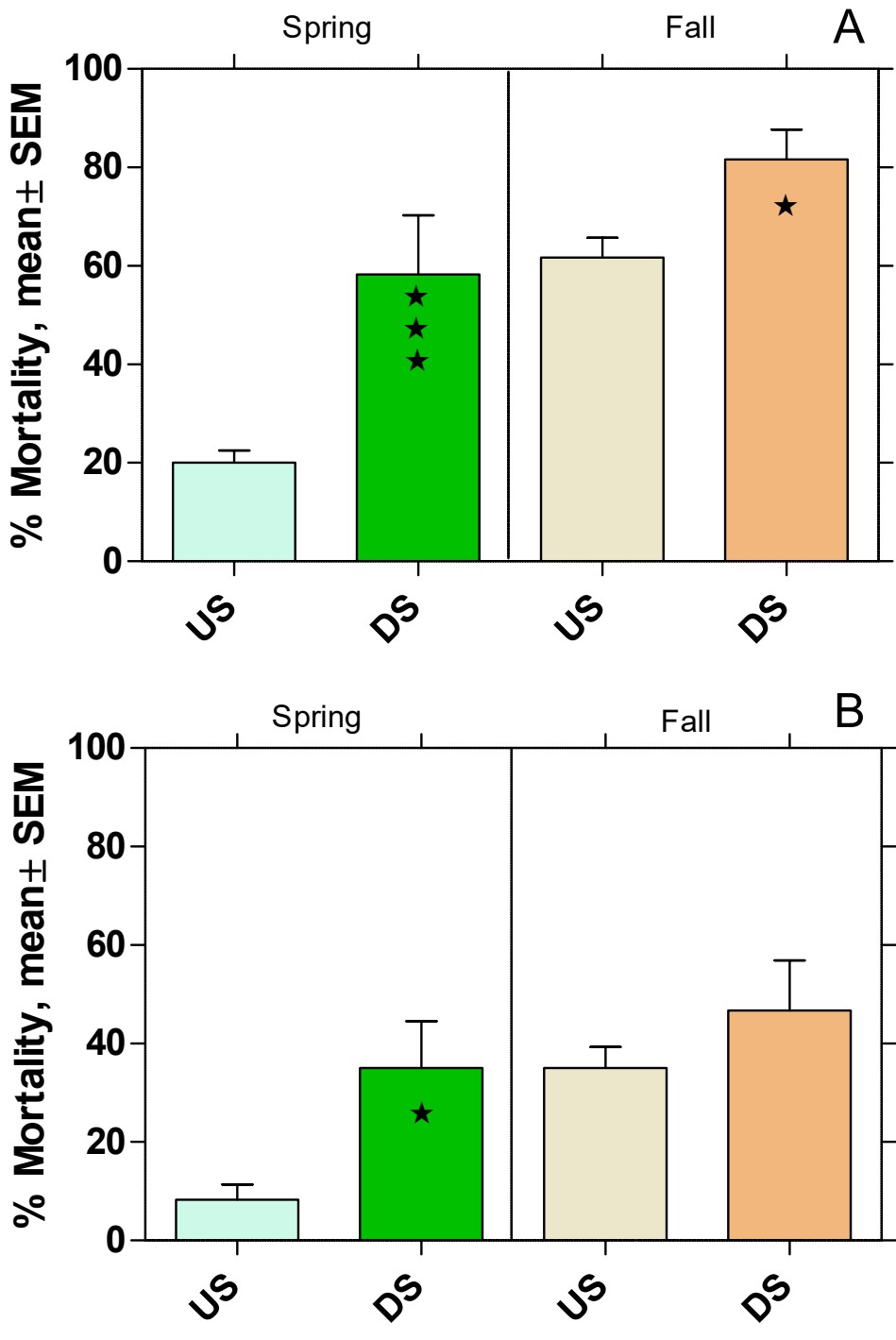

**Figure 6 Mortality of *Gammarus fossarum* in the active monitoring campaign over one month in the upstream (US) and downstream (DS) site in the receiving rivers of the WWTPs Mörfelden-Walldorf (A) and Bickenbach (B).** Significant differences refer to DS compared to US (Fisher's exact test; ★ $p < 0.05$, ★★★ $p < 0.001$).

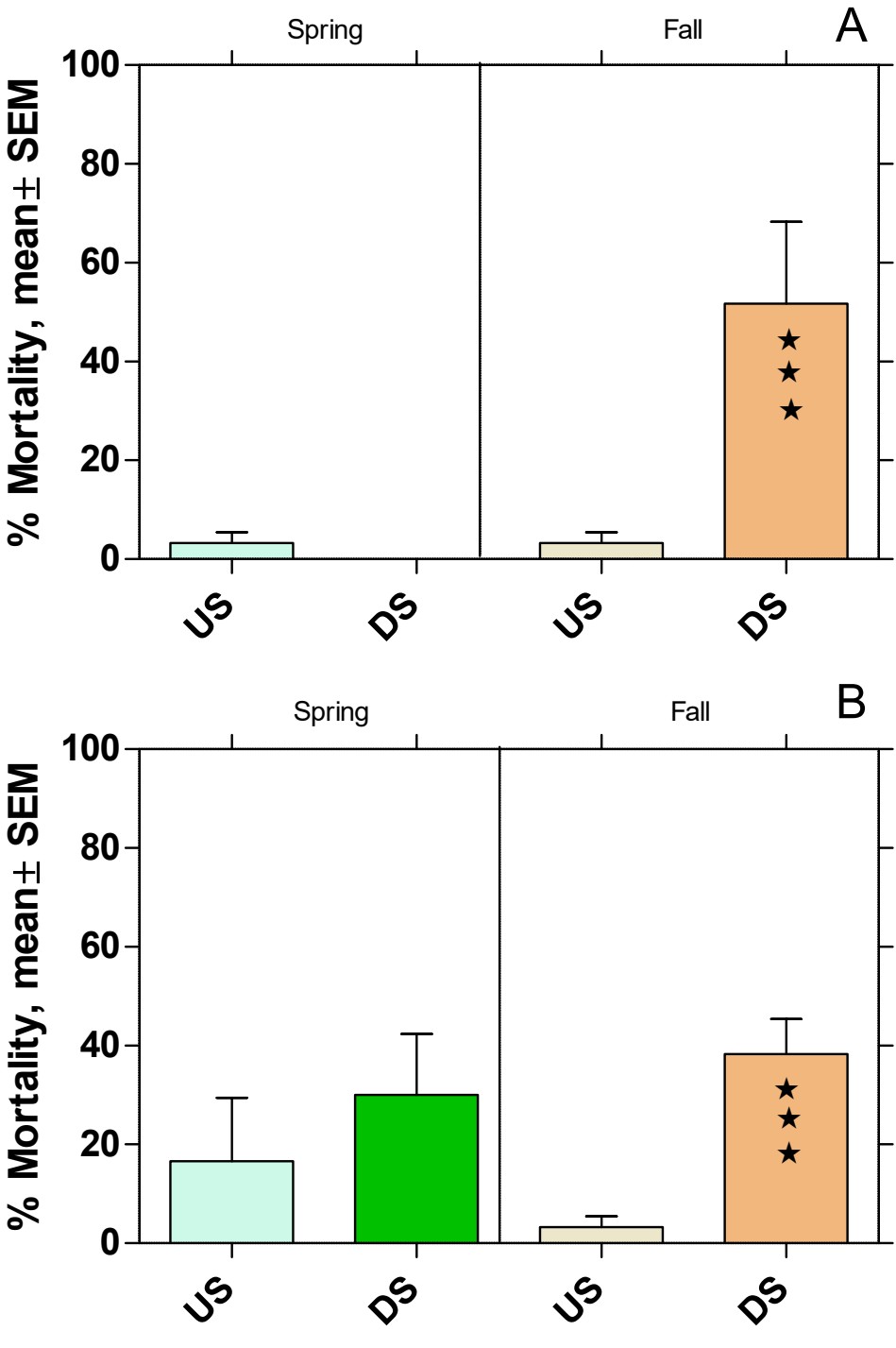

**Figure 7** Mortality of *Potamopyrgus antipodarum* in the active monitoring campaign over one month in the upstream (US) and downstream (DS) site in the receiving rivers of the WWTPs Mörfelden-Walldorf (A) and Bickenbach (B). Significant differences refer to DS compared to US (Fisher's exact test; ★★★ $p < 0.001$).

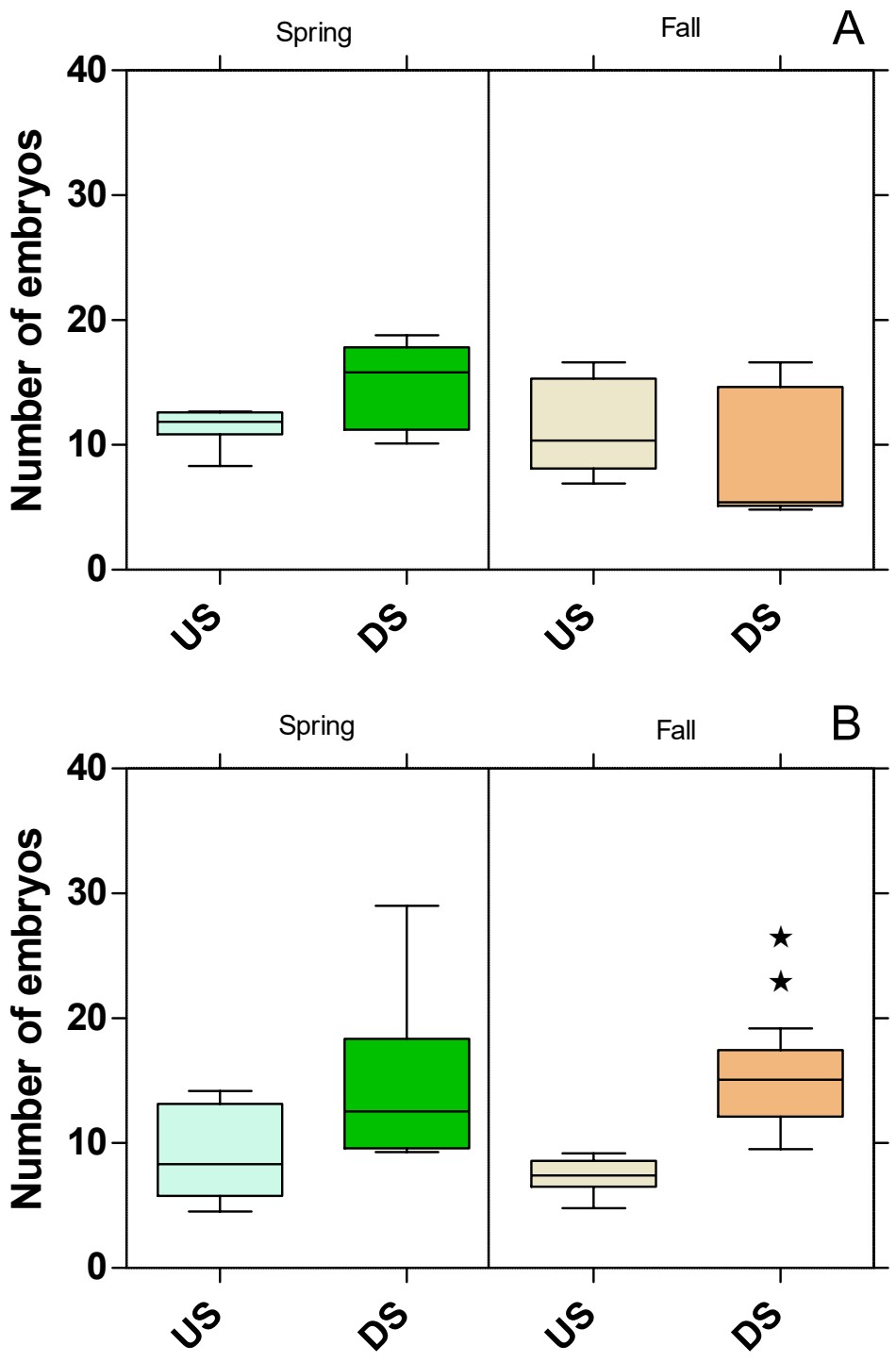

**Figure 8** **Number of embryos in *Potamopyrgus antipodarum* in the active monitoring campaign over one month in the upstream (US) and downstream (DS) site in the receiving rivers of the WWTPs Mörfelden-Walldorf (A) and Bickenbach (B).** Significant differences refer to DS compared to US (unpaired *t*-test with Welch's correction for unequal variances; ★★ $p < 0.01$).

## DISCUSSION

### Physicochemical parameters

Regardless of the effectiveness of both WWTPs in reducing the concentrations of phosphate and inorganic nitrogen, they still had a strong impact on the physicochemical parameters in the receiving streams, which carried between 78% and 92% treated WW downstream of the discharger. The oxygen saturation decreased in DS-MW and DS-B during both seasons. The conductivity and the nitrate concentrations exceeded the reference value of 800 $\mu$S/cm and 0.05 mg/L, respectively for river type 19 in all DS samples during the spring and fall campaign (*Halle & Müller, 2014*). Despite the efficient ammonium removal in both WWTPs, the ammonium concentrations in the DS samples exceeded the reference value of 0.4 mg/L by up to factor 2.8 (MW in spring). Conversely, the concentrations of nitrate and phosphate were below the reference value during spring and fall in the DS samples (10 and 0.05 mg/L, respectively) (*German Environment Agency, 2017*; *Halle & Müller, 2014*). Finally, the mean values of the pH (not shown in Table 3) have met the requirements for good chemical status.

### *Baseline toxicity*

The bioluminescence of *Aliivibrio fischeri* is directly proportional to their metabolic activity, so any disruption by toxic substances results in a decreased luminescence. Considering this, a marked inhibition of the bioluminescence by the raw WW (MT) in both WWTPs was observed ($EC_{50} < 6.45 \pm 4.16$ REF). This observation is in accordance with previous reports (*Macova et al., 2011*; *Völker et al., 2017*). Both conventional WWTPs in B and MW eliminated the baseline toxicity of the raw WW during biological treatment by more than 90%. This elimination performance is also comparable to the findings of other studies, which report elimination rates between 86% and 94% (*Völker et al., 2017*; *Yu et al., 2014*).

Despite the considerable elimination of baseline toxicity during the WW treatment process in both WWTPs, the effluents still had a major impact on the receiving waterbodies, so that the downstream sites during the spring and fall campaign exceeded the non-toxic limit in the Microtox assay. These results suggest that the level of baseline toxicity was higher during spring than in the fall campaign in both WWTPs, which is in line with previous studies that concluded that the toxicity of effluents was usually higher in spring and summer (*Vasquez & Fatta-Kassinos, 2013*). This can be explained by the main application of plant protection products on agricultural fields in spring. Increased runoff can wash pesticides into water bodies and contribute to higher activity in the Microtox assay, which is a bioassay that detects the non-specific toxicity of a broad range of substances (*Tang et al., 2013*). This is in contrast to *in vitro* assays for endocrine, dioxin-like and mutagenic activity, which are specific for a particular mode of toxicological action (*Wagner et al., 2017*).

### *Endocrine activity*

The endocrine profile of the influent samples indicates that WW contains compounds with estrogenic, androgenic, and anti-estrogenic activities. Our finding of an elimination of estrogenic and androgenic activities by more than 90% in the conventional treatment

Trejos Delgado et al. (2024), *PeerJ*, DOI 10.7717/peerj.17326

Peer J

**Table 3 Physicochemical parameters at the sampling sites.** Mean values of physicochemical parameters during the spring and fall campaign. All data were measured on two occasions (4 weeks difference) during each campaign. BT-L data during the spring campaign was not measured.

| Parameter | Season | Sample sites | | | | | | | | |
|---|---|---|---|---|---|---|---|---|---|---|
| | | BT-L | US-MW | MT-MW | BT-MW | DS-MW | US-B | MT-B | BT-B | DS-B |
| Oxygen saturation[%] | Spring | – | 99.1 | 65.0 | 68.5 | 78.7 | 92.6 | 9.00 | 82.5 | 65.0 |
| | Fall | 104 | 83.2 | 20.0 | 93.6 | 77.7 | 85.8 | 58.3 | 61.2 | 68.1 |
| Conductivity [$\mu$S/cm] | Spring | – | 781 | 1,190 | 772 | 817 | 493 | 1,560 | 952 | 967 |
| | Fall | 1,080 | 976 | 1,270 | 934 | 1,000 | 532 | 1,500 | 1,040 | 932 |
| Phosphate P [$PO_4^{3-}$-P mg/L] | Spring | | 0.04 | 1.44 | 0.04 | 0.03 | 0.05 | 0.68 | 0.02 | 0.02 |
| | Fall | 0.03 | 0.03 | 0.87 | 0.04 | 0.02 | 0.02 | 0.67 | 0.02 | 0.02 |
| Ammonium-N [$NH_4^+$-N mg/L] | Spring | | 0.03 | 34.5 | 1.81 | 1.11 | 0.02 | 52.5 | 0.26 | 0.15 |
| | Fall | 0.61 | 0.19 | 44.7 | 0.05 | 0.18 | 0.11 | 23.2 | 0.29 | 0.32 |
| Nitrite-N [$NO_2^-$-N mg/L] | Spring | | 0.02 | 0.07 | 0.06 | 0.07 | 0.02 | 0.03 | 0.10 | 0.03 |
| | Fall | 0.05 | 0.05 | 0.04 | 0.01 | 0.03 | 0.02 | 0.04 | 0.04 | 0.04 |
| Nitrate-N [$NO_{3-}$-N mg/L] | Spring | | 1.35 | 0.16 | 1.53 | 1.08 | 0.45 | 0.46 | 1.13 | 1.03 |
| | Fall | 2.21 | 1.72 | 0.13 | 2.55 | 1.78 | 0.51 | 0.32 | 0.93 | 0.67 |

process is in line with the results of other studies which report an elimination between 88% and 95% (*Archer et al., 2020*; *Arlos et al., 2018*; *Giebner et al., 2018*). In the case of anti-estrogenic activity, the efficiency at the WWTPs MW (above 80%) and B (above 90%) was even higher compared to the results of *Giebner et al. (2018)* reporting a reduction of anti-estrogenic activity during conventional treatment of only 15%.

Although the estrogenic activity of the WW was reduced by more than 90% in both WWTPs, the residual activity was sufficient to adversely affect the receiving waters. This is particularly noticeable at the Landbach, the receiving water of WWTP B, where the estrogenic activity increased most significantly downstream of the WWTP ($p < 0.001$). At the Geräthsbach, the estrogenic activity increased significantly only in spring downstream of the WWTP MW ($p < 0.01$) and in fall it even decreased significantly ($p < 0.05$). These results are in line with results in a survey of 36 Danish WWTPs and their effluents (*Danish Ministry of the Environment, 2005*). In almost 70% of the water courses receiving effluents from WWTPs, the level of estrogenic activity downstream the discharge point was higher than the activity upstream of the WWTP effluent. Also, in the study of *Aerni et al. (2004)* who compared the estrogenic effect upstream and downstream of five Swiss and French WWTPs support our findings. At all upstream sampling sites, the estrogenic activity was below the limit of quantification, while in the effluent the activity was on average between 0.4–5.5 ng E-EQ/L.

The reason for the lower impact of the WWTP MW on the estrogenic activity in the Geräthsbach compared to the WWTP B and the Landbach is that the Geräthsbach is already influenced by the WWTP Langen upstream of WWTP MW. Accordingly, the WWTP MW is not the only and first source of contamination. This pattern was observed during the spring sampling campaign for almost all analyzed activities (baseline toxicity, estrogenic and dioxin-like activity, mutagenicity). The WWTP Langen is located 10 km upstream of MW and is with 54,600 population equivalents even larger than MW (*Hessian Ministry for the Environment, Climate Protection, Agriculture and Consumer Protection (HMUKLV), 2018b*). For this reason, we decided to take an additional sample at the effluent of WWTP Langen during the fall campaign. The result for the estrogenic activity supports the assumption that the effluent from the WWTP Langen is responsible for the higher activity in the US site of the Geräthsbach compared to the Landbach.

We observed that the estrogenic activity in both WWTPs effluents was significantly higher during the fall campaign than in spring ($p < 0.001$). Also, this finding is in line with the Danish survey at 36 WWTPs which reports the highest levels of estrogenic activity in summer and fall and the lowest during winter and spring (*Danish Ministry of the Environment, 2005*). A possible explanation is the higher precipitation in winter and spring so that the concentrations of estrogenically active compounds in the streams are lower than in the dry summer and fall. The mean flow (MQ) of 238 L/s in the Geräthsbach and 125 L/s in the Landbach according to *Hessisches Landesamt für Naturschutz, Umwelt und Geologie (HLNUC) (2022)* was slightly exceeded during our spring campaign while it was only 175 L/s and 96 L/s in the two rivers during the fall campaign. Furthermore, phyto-estrogenic material is introduced *via* fallen leaves (*Janeczko, 2021*), increasing the estrogenic load in the water bodies.

The lack of a measurable androgenic and anti-androgenic activity is supported by previous studies (*Brzezinska, Sakson & Olejnik, 2023*; *Li et al., 2010*). The same was observed for the anti-estrogenic activity, where no significant difference was found between upstream and downstream sample points (values close to the LOD = 1.84 mg OHT-EQ/L). This finding contrasts the report by *Brzezinska, Sakson & Olejnik (2023)* where an antagonistic activity up to100 mg OHT/L was found in the effluent of the WWTP Lodz in Poland. Despite the negative findings in this project phase, the three *in vitro* assays will be used as EBM even after the 4th treatment stage has been implemented, since, for example, anti-estrogenic activities can arise in the wastewater during ozonation (*Schneider et al., 2020*; *Stalter et al., 2011*).

## Dioxin-like activity

The raw WW at the WWTP MW contained compounds with aryl-hydrocarbon receptor agonistic activity, resulting in a high dioxin-like activity, which was reduced by more than 50% during biological treatment. This elimination rate is lower than in the study of *Magdeburg et al. (2014)* reporting an elimination of dioxin-like activity of more than 81%. For WWTP B, the elimination of dioxin-like activity through conventional WW treatment could not be determined because the raw WW samples were cytotoxic after mechanical treatment or the enzymes responsible for luminescence were directly inhibited by the samples (*Reifferscheid et al., 2011*). *Völker et al. (2016)* found a similar result for the raw WW of a WWTP in their study.

Despite the significant reduction of dioxin-like activity in both WWTPs, the remaining activity was sufficient to significantly increase the activity in the Geräthsbach and Landbach, except for the fall sample from the Geräthsbach, where the activity increased by 12.6% which was not significantly different. The measured activities in both rivers are comparable to the rivers Horloff (up to 1150 ng β-NF-EQ/L) and Nidda (up to 1,180 ng β-NF-EQ/L) in the north of Frankfurt, which also receive effluents form WWTPs but higher compared to rivers not impacted by WWTP effluents with activities less than 200 ng β-NF-EQ/L (*Brettschneider et al., 2019b*; *Brettschneider et al., 2023*).

The seasonal comparison shows significantly higher dioxin-like activities ($p < 0.001$) at all sampling points in MW in spring compared to fall, while for the WWTP B there is only a corresponding trend without significant differences. Although both WWTPs contribute to the high dioxin-like activity in the rivers, other sources are relevant. The dioxin-like activity in the water phase and in the sediments of rivers is primarily caused by polycyclic aromatic hydrocarbons (PAHs), which enter the waters also *via* atmospheric deposition and surface runoff from traffic areas (*Boxall & Maltby, 1997*; *Maltby et al., 1995*).

## Mutagenicity (Ames fluctuation test)

The findings of high mutagenic activities in samples treated with S9 mix show that the conventionally treated WW contains substances that are only converted into mutagenic compounds through metabolic activation, as typically occurs by enzymes in the liver of vertebrates or the midgut gland of molluscs, crustaceans and other invertebrate taxa. Furthermore, it shows that conventional WW treatment is not sufficient to eliminate

these compounds. The high mutagenic activity at the sampling points in both rivers influenced by the discharge of treated WW demonstrates the urgent need for action to reduce this undesirable effect through advanced WW treatment. *Giebner et al. (2018)* were able to show that activated carbon filtration in particular is suitable for removing mutagenic substances from WW, while oxidative processes such as ozonation can lead to the formation of alkylating agents, nitrosamines and other mutagenic substances in the treated WW (*Mestankova et al., 2014*; *Schmidt & Brauch, 2008*; *Tsutsumi, Inami & Mochizuki, 2010*).

## Active monitoring with *Gammarus fossarum* and *Potamopyrgus antipodarum*

To investigate potential pollution effects on the invertebrate community, we conducted an active monitoring study with the crustacean *Gammarus fossarum* and the gastropod *Potamopyrgus antipodarum*. Both species have widely been used for ecotoxicological studies, including the assessment of WWTP effluents and their impacts on the biocenosis in receiving waters (*Brettschneider et al., 2019a*; *Brettschneider et al., 2023*; *Harth et al., 2018*; *Schneider et al., 2020*). *G. fossarum* is an indicator species of a low-mountain range biocenosis (*Landesamt fuer Natur Umwelt und Verbraucherschutz (LANUV), 2015*), is particularly sensitive to anthropogenic pressures on water bodies and plays an important role in the food web as a shredder and food for macroinvertebrates, fish, or birds (*Besse et al., 2013*). This dioecious and oviparous species has a breeding season of up to 10 months per year, with reproduction peaking in late winter and spring. Female *G. fossarum* retain the fertilized eggs and developing embryos until hatching in an open brood chamber, the marsupium, which is formed by oostegites, long appendages of the gill-carrying legs. Although changes in sex ratio and feminization effects have been described in freshwater amphipods following exposure to synthetic estrogens (*Watts, Pascoe & Carroll, 2002*) and estrogenically active WW (*Schneider, Oehlmann & Oetken, 2015*), the typical response of *G. fossarum* to exposure to chemicals is a decrease of the number of eggs/embryos in the marsupium (*Brettschneider et al., 2019a*; *Brettschneider et al., 2023*; *Harth et al., 2018*). *P. antipodarum* is an euryoecious snail inhabiting small creeks, streams, lakes, and estuaries. Males are extremely rare in European populations and were never found in our long-term laboratory culture and their source populations with more than 200,000 specimens analyzed. Females reproduce parthenogenetically and are ovoviviparous: Eggs develop in the anterior part of the pallial oviduct section, which is transformed into a brood pouch. Older embryos are situated in the anterior and younger embryos in the posterior part of the brood pouch. The embryos are released through the female aperture when the eggshell tears open (*Organisation for Economic Co-operation and Development (OECD), 2016*). As with *G. fossarum*, this allows to assess the reproductive output of each female by evaluating the numbers of eggs and embryos in the marsupium or brood pouch. *P. antipodarum* is particularly sensitive to endocrine active chemicals and responds to exposure to estrogenic substances by increasing embryo production (*Duft et al., 2003*; *Duft et al., 2007*; *Jobling et al., 2004*).

Given the prominent ecological function of *G. fossarum* in the aquatic community of low-mountain range rivers, the significant increased mortality downstream of the WWTP MW to 59% in spring and 81% in fall is worrying. Although the same tendency was observed downstream of the WWTP B in both seasons, the mortality levels were lower and increased only significantly during the spring campaign. These finding are in line with results of other active monitoring studies such as *Brettschneider et al. (2019a)*, who found increased mortality rates of 42% and 68% in *G. fossarum* downstream of WWTPs with 43,800 PE and 78,000 PE, respectively.

Also, the mortality of *P. antipodarum* increased significantly downstream of both WWTPs in fall, while there was no significant change in mortality in spring. This finding corresponds to previous studies on other WW-influenced rivers, in which a significant increase in the mortality of *P. antipodarum* occurred downstream of WWTPs, especially in fall, while the effects were significantly smaller in spring (*Brettschneider et al., 2019a*). In addition, in previous studies in which both species were used together for active monitoring, mortality was generally lower for *P. antipodarum* than for *G. fossarum*, possibly because the snail can avoid exposure peaks by temporarily closing its shell with the operculum (*Brettschneider et al., 2023*).

While no influence on reproduction could be determined for *G. fossarum*, the number of embryos for *P. antipodarum* increased downstream of the two WWTPs, except for the fall campaign in the Landbach. Since it is known that *P. antipodarum* responds to exposure to substances with estrogenic activity by increasing embryo production (*Duft et al., 2003*; *Duft et al., 2007*; *Jobling et al., 2004*) and the estrogenic activity determined by the YES was significantly increased at the study sites located downstream the WWTPs, this indicates that the estrogenic exposure is the responsible factor for the higher embryo numbers. The very high toxicity below the WWTP MW in fall, which is also reflected in the highest mortalities in *G. fossarum* and *P. antipodarum*, possibly masks the estrogenic effect on the number of embryos in the snail. Also in other studies, the number of embryos in *P. antipodarum* increased below WWTP effluents or when exposed to conventionally treated WW (*Brettschneider et al., 2019a*; *Schneider et al., 2020*; *Stalter, Magdeburg & Oehlmann, 2010*).

Together, these findings demonstrate that the applied methods are suited to assess the negative effects of effluents from conventional WWTPs on the receiving water bodies and the aquatic biocenosis in these rivers. For the next project phase after the implementation of the 4th treatment step with a combination of ozonation, followed by activated carbon filtration in the two WWTPs, the same physicochemical parameters and EBMs should therefore be used to enable an effective comparison between the status quo before and after implementation. However, the *Salmonella typhimurium* strain YG7108, which has proven to be particularly sensitive for detecting mutagenic compounds formed during ozonation (*Giebner et al., 2018*; *Magdeburg et al., 2014*; *Schneider et al., 2020*), will also be considered for the Ames fluctuation test in the future. This makes it possible to check whether these compounds are effectively removed by activated carbon filtration after ozonation. The use of the same battery of EBMs allows to assess the efficiency of the upgrading measure and to record potential pollution reduction in the receiving waters.

The findings of our study can be extrapolated to other smaller rivers that receive conventionally treated WW from relatively large WWTPs. The current study demonstrates that the effluents originating from the two WWTPs negatively affect key life cycle of representative macroinvertebrate species in the water bodies and underlines the necessity to upgrade WWTPs with a 4th treatment step to achieve the target of a good chemical and ecological status of surface waters according to the EU Water Framework Directive (*European Commission, 2019*).

## CONCLUSIONS

The findings of this study showed that EBMs are an efficient tool to analyze the multiple effects that complex mixtures, such as WW, have in the aquatic environment. These methods revealed that although both WWTPs had high toxicity removal rates, remaining baseline toxicity, endocrine activity (especially estrogenic activity) and dioxin-like activity, had a negative impact on both streams. In the specific case of mutagenic effects, high activities were found in both WWTPs, where the effluents increased the mutagenic activity compared with the samples taken upstream.

All *in vitro* assays used in our study were sufficiently sensitive to assess the respective activity levels in conventionally treated WW and in the receiving streams, including seasonal differences, allowing an integrative impact assessment of the effluents form both WWTPs. The differences between spring and fall seem to be related to the different precipitation conditions and the resulting different dilution of WW. The active monitoring with *Gammarus fossarum* and *Potamopyrgus antipodarum* also showed higher effect levels on mortality (both species) and reproduction (*P. antipodarum*) during the fall campaign. The *in vivo* tests with both species thus confirm the findings of the *in vitro* assays at a biologically higher and ecologically more relevant level.

In summary, both hypotheses are supported by our findings: conventionally treated WW showed a wide range of activities in assays addressing specific modes of toxicological action but also a high baseline toxicity. These effects were also measured in the receiving surface waters downstream the WWTPs. Therefore, our study demonstrated that conventional WW treatment is insufficient to prevent negative impacts on the receiving aquatic ecosystem. This implies that further advanced WW treatment is urgently needed to guarantee safe effluents in these streams.

### Funding

Catalina Trejos Delgado received a scholarship from COLFUTURO-DAAD. The funders had no role in study design, data collection and analysis, decision to publish, or preparation of the manuscript.

### Grant Disclosures

The following grant information was disclosed by the authors:
COLFUTURO-DAAD.
## Competing Interests

Jörg Oehlmann is an Academic Editor for PeerJ.

## Author Contributions

- Catalina Trejos Delgado conceived and designed the experiments, performed the experiments, analyzed the data, prepared figures and/or tables, authored or reviewed drafts of the article, and approved the final draft.
- Andrea Dombrowski performed the experiments, analyzed the data, authored or reviewed drafts of the article, and approved the final draft.
- Jörg Oehlmann conceived and designed the experiments, analyzed the data, authored or reviewed drafts of the article, and approved the final draft.

## Data Availability

The raw measurements are available in the Supplementary File.

## Supplemental Information

Supplemental information for this article can be found online at http://dx.doi.org/10.7717/peerj.17326#supplemental-information.

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
