# Peer review of "Assessing the impact of two conventional wastewater treatment plants on small streams with effect-based methods"

_PeerJ, doi:10.7717/peerj.17326_

## Round 0.1 · original submission · Minor Revisions

Three Reviewers have provided their opinions and comments regarding your manuscript. The Reviewers are all in favor of publishing the manuscript in PeerJ and consider the presented results a valuable contribution to the field of effect-based hazard assessment.

The Reviewers have made a number of suggestions for modifying the manuscript. Please address these in the manuscript and your response letter if you agree these changes improve the manuscript. Otherwise, please explain in your response any unaddressed Reviewer comments.

**Language Note:** The review process has identified that the English language must be improved. PeerJ can provide language editing services - please contact us at copyediting@peerj.com for pricing (be sure to provide your manuscript number and title). Alternatively, you should make your own arrangements to improve the language quality and provide details in your response letter. – PeerJ Staff

Reviewer 1 ·

Basic reporting

The manuscript titled: “Assessing the impact of two conventional wastewater 2 treatment plants on small streams with effect-based 3 method” by Trejos-Delgado et al., had the main objective to determine baseline effects of two municipal wastewater treatment plant effluents on their respective receiving systems (over 2 seasons) before the upgrade implementation of the advanced 4th step treatment consisting of ozonation followed by activated carbon filtration. Relying on effect-based methods, the authors used a battery of bioassays to assess baseline non-specific toxicity, mutagenicity, endocrine- and dioxin-like activities, and effects on survival and reproduction on a gammarid species and a mud snail species. The manuscript is well written, and it reads relatively smoothly. The objectives are well defined, supported with the appropriate methodology and stats, and they are addressed both in the results and discussion sections in a consistent order.

Experimental design

In terms of experimental design adequacy, selection of tests used and statistical analyses the authors did a good job and I have no concerns regarding these. The quality of tables and figures is satisfying, there are minor suggestions on clarity improvement in the specific comments section. Further, there was a few concerns that I had regarding clarity, mostly in the materials and methods section, less in the results and discussion. These are addressed in detail in the specific comments section of this review.

Validity of the findings

Even though the research is not particularly innovative, it is important because it establishes the baseline effects as it examines the pre-upgrade conditions in 2 receiving environments and it sets up the stage for the follow up (post-upgrade) studies that will determine the effects of the upgraded treatment on the receiving environments. The results of the current study clearly indicate the necessity of an advanced treatment implementation and they indicate that the conventional wastewater treatment is insufficient to maintain a healthy status for the respective aquatic environments, their resident biota but it also suggests potential risk for human health.
I think that the current manuscript should be taken into consideration for publication in the PeerJ journal after the authors address the suggestions and comments provided in this review.

Additional comments

Specific comments:
Line 90: Specify that the manuscript is looking at the current pre-upgrade conditions or conventional municipal wastewater treatment on the receiving aquatic systems. Initially it is a bit unclear whether the study encompasses both pre- and post-upgrade effects.
Emphasize that the focus is on establishing the baseline so that the follow-up studies can determine the effects upon addition of more advanced wastewater treatment process.
Line 97: “…water downstream the WWTP effluents – I suggest removing “effluents”, it is downstream of the plants or outfalls.
Line 125: specify if these were samples of surface water collected upstream/downstream of the outfalls or treated effluent samples collected from the plants? Table 1 lists the sites/locations, but it would be helpful if this was addressed in the respective section as well (technically, there are 4 types of samples…) – consider using the site codes to better describe the sample types: surface US and DS and WW mechanical and biological treatment.

Line 127: Please specify if two samples per season per site were collected.
Line 131: “…chemical parameters such as …” if the listed chemical parameters were the only parameters measured then instead of using such as I suggest using the following word: including.
If more than the listed parameters were measured, then please specify all. The phrase “such as” implies that there were more parameters measured but for some reason not all of them are listed.
Line 141: 2000 mL vs 2 L (line 124) please be consistent in terms of volume units - either stick with mL or L.
Line 179: what are native samples? Like filtered surface water samples? Earlier it was described that 2000 mL (or 2L) sample was filtered and SPE processed but there is no mention of “native sample” leftover for the YAES and YAAS assays. Consider adding that to the methodology earlier
Line 219: change samplings sites to sampling sites
Line 217: I assume that the upper reach of the Urselbach River was chosen because it is an appropriate/optimal reference or control site that has minimum anthropogenic effects – clarify (if there are any studies supporting this, please refer to them).
Where were the gammarids kept before the onset of the experiment for one week? Were they collected at the Urselbach site and kept in enclosures at the same site – specify.
Lines 236/237: same comment as above (line 217) – were these sites considered reference/control because of minimum anthropogenic impact or minimum disturbance.
Line 257: was alpha 0.005 or 0.05, the first seams to be very restrictive but if that was the case can you explain the reason of setting such a stringent test.
Lines 257/258: arrows pointing down versus star symbols in the figures indicating significant difference
Results section: Lines 264-284 – consider using the sampling site codes defined in table 1 when describing your findings rather then just referring sporadically to treatment steps or the receiving streams – it is easier for the reader to follow if these codes are added in the results section than to go back and forth between the text and the table content.
In the discussion part you should consider adding a tentative plan for the follow-up studies that will look at the post-upgrade conditions; what implications would these have for the similar streams in Germany or Europe and municipal wastewater treatment plants that are still using the conventional wastewater treatment process and are not meeting quality standards.
Lines 398/399: Would the use of more specific bioassay be able to discriminate between toxicity originating from the municipal wastewater effluents versus surrounding areas – would the same bioassay be used in the follow-up studies?
Line 444/445: Can you include stream flow data for fall and spring or perhaps reference an earlier study that looked at seasonal changes in the flow?
Lines 449-454: are these bioassays relevant for the future studies – elaborate why yes or no?
Lines 535-546: I suggest adding a paragraph on explaining the main reproductive strategies of the gammarid and the snail used in the study, as well as the potential effects of endocrine disrupting compounds on their reproduction – consider looking at whether these species have estrogen receptors. Is there any literature on mechanisms of action of estrogens on increased number of embryos in mud snails downstream wastewater treatment plants.
Conclusions: Consider reiterating seasonal changes between the two wastewater plants since this was one of the objectives of the study – provide an explanation between bioassays differences and sensitivities, their relevance for the respective seasons and alternative options.

***autumn or fall – use one for consistency

Figure 1 caption – consider adding a more detailed explanation of the figure – it is obvious that the figure is a map of the sites but the caption should contain similar information that was provided in table 1 – perhaps combine these, especially add the site codes to the map.

·

Basic reporting

The manuscript “Assessing the impact of two conventional wastewater treatment plants on small streams with effect-based methods (#92111)” is written in correct English and uses technically precise terminology. The text is flowing and easy to read.
All the citations reported along the text are well appropriate and quite updated.
Relevant literature, regarding the presence of micropollutants in WWTPs, is appropriately referenced in the first part of the introduction. And, also the findings of the work are widely supported by appropriate literature. However, the implementation of EBMs should be better referred within the Introduction (Line 89), also because EBMs are the core of the paper. I suggest to briefly introduce the Effect based Methods (what are) and their usefulness for detecting the mixture effect, for identifying the drives of toxicity and so on. It would be worth adding some literature such as Bioanalytical Tools in Water Quality Assessment by Escher et al., 2021 or The European technical report on aquatic effect‐based monitoring tools under the water framework directive. Environmental Sciences Europe, 27(1), Article 7. https://doi.org/10.1186/s12302-015-0039-4.
All Figures along the manuscript are appropriately described and labeled. The Authors chose to represent the results in a way that is very easy to understand. All the results fit perfectly with the initial hypothesis.

Experimental design

Line 90: The results of this work describe the situation only before the implementation of the 4th treatment. I understand that in a second step, after the implementation of the WWTP treatments, a second study will be conducted. Please clarify the sentence describing the aim of the present work (see comments in pdf file).

Validity of the findings

The reported data are statistically sound and controlled. No comments nor suggestions are needed for the results. Conclusions are well stated, linked to original research hypotheses and limited to supporting results.

Additional comments

This is a very attracting paper. I appreciate so much the topic of the use of the EBMs for river waters affected by WWTPs. It is a topical issue. I admire as the Authors presented the in vitro tests. The adoption of this bioassays is at the cutting-edge of water ecotoxicology.
The use of so many tests within an ecotoxicological battery is very impressive and this aspect could even have been emphasized more. The selection of the adopted tests and the discussion of the results are in line with the main objective of the work, that is to describe the current status of toxicity level of WW and water courses WW-affected before the implementation of treatments.
For the future, it could be very interesting to perform parallel chemical analyses in order to deeply study the complex mixture of WW and and to possibly identify the main drivers of toxicity.

Reviewer 3 ·

Basic reporting

1.1. The Abstract: The author (s) should provide (i) The purposes/the values of study in the study aim, (ii) Short describe about the methods, variables, hypotheses and how the results support the hypotheses.
1.2. The Introduction: The author (s) should check gramma and spelling, appropriate tense and words using for academic writing.
1.3. The Methods: (i) The author (s) should check gramma and spelling, appropriate tense (should be the past tense for work had been done in the past) and words using for academic writing. (ii) The method should be rearranged: Collect sample > Store sample > Pretreatment sample > Analysis ... (iii) the study includes sampling/Monitoring sites, therefore should use ANOVA and Turkey post host test to show the different in the findings.(iv) The measurements (data, figures) by microscope should provide in Appendix
1.4. The Results: The author (s) should rearrange the results by provide the works you did and the results you obtain, remove comparisons with the other publications.
1.5. The Discussion: The author (s) should not present results here, but short mention about the key results and then compares to the similar studies, explain your finding, limitation and recommendation.
1.6. The Table: The author (s) should provide clearly about the Legend for each table, consistent in presenting the numbers, "n.a", "-", "+)...
1.7.The Figure: The author (s) should provide clearly about the Legend for each Figures. The way to put “*” on the Figure.
1.8. The author (s) should check for consistent in using "," before "and", tense, ...overall gramma and spelling for the whole manuscript.

Experimental design

2.1. The methods should be rearranged logically.
2.2. Statistical method should be reconsidered.

Validity of the findings

3.1. The author (s) should provide additional data, figure, statistical output. in to the Appendix.

---

## Round 0.2 · Minor Revisions

Dear authors,

Please address the additional comments of Reviewer 1.

**Language Note:** The review process has identified that the English language must be improved. PeerJ can provide language editing services - please contact us at copyediting@peerj.com for pricing (be sure to provide your manuscript number and title). Alternatively, you should make your own arrangements to improve the language quality and provide details in your response letter. – PeerJ Staff

Reviewer 1 ·

Basic reporting

Thank you for giving me the opportunity to review the manuscript titled: “Assessing the impact of two conventional wastewater treatment plants on small streams with effect-based methods” by Trejos-Delgado et al., and to potentially contribute both to Peer J and the scientific community that this journal targets.
I read the revised manuscript carefully and I have no concerns regarding the methodology and the experimental design used, the authors added clarity where reviewers identified the need for. I believe that the authors addressed reviewers’ comments and concerns to the best of their knowledge and abilities, and I believe that the revised manuscript is significantly improved. Although the authors made some changes that improved the language style and grammar, I have an impression that there is still room for improvement. I suggest the authors use the Peer J language editing services to improve the writing style and make it more concise and more “native speaking English-like”.

Experimental design

no concerns

Validity of the findings

no concerns, specified in the initial reviewers' response

Additional comments

I noticed that the results are presented in a specific way, namely each result starts with an identifier for the figure/table where the respective result is visually graphed, or values are assigned. I am not sure if this is a journal style requirement or the authors preference. If first, then ignore comment but if second, then consider removing all those introductory sentences that specify figure/table location and just refer to the figures/tables by using parentheses. This will make things more concise and easier to follow. I hope this makes sense.


As I previously indicated, the original manuscript required minor revisions and I think that that the revised manuscript still upholds that status (minor rev), but some additional work is required before the manuscript is fully accepted for publication in Peer J.

The specific comments are listed below.



Line 68: instead of were implemented consider using “were designed”

Lines 74-78: This sentence does not read well: the appositive is too long and distractive, consider modifying it and make it more concise.
Lines 88-80: repetitive, pharmaceuticals and pesticides were listed earlier as micropollutants in line 58.
Lines 84-89: great that you added more information on selected micropollutants and their respective concentrations, but the sentence is too long now – consider splitting it into at least two.
Line 91: the construct “micropollutant contamination” overemphasises the subject in the sentence; the noun “micropollutant” already implies that these substances pollute or contaminate the environment; consider replacing “contamination” with “concentration”.
Line 114: there is a colon after “bioassays” followed by an upper-case letter – is that on purpose? Also consider numbering the different assays used (e.g., (1)….(2)…).

Lines 131-133: I think the use of articles (“a”) is unnecessary exhibit poor quality….as well as moderate …strongly (highly altered sounds better?) or completely altered …and poor ecological status…

Line 155: use “bringing” instead of “returning” the samples
Line 317: consider deleting the first sentence
Lines 317-319: Consider re-writing the sentence: The baseline toxicity, determined with the Microtox assay, was significantly higher DS compared to US, except at MW MWWTP in fall (Figure 2).
Line 323: “basic” vs “baseline” toxicity?
Lines 330-333: Consider re-writing: “The estrogenic activity increased significantly at the DS sites compared to US at both WWTPs, with the exception of MW WWTP in fall which showed a 6% decrease (Figure 3).” – simplify the sentences and remove unnecessary explanations, like the first sentence (329), in this paragraph.

Lines 333-336: same comment as above: “Despite over 90% removal by WWTPs, estrogenic activity was 22% higher at the MW WWTP in spring, and 60% at the B WWTP in both seasons compared to their respective US sites.”
Line 337: Change the order of words: “The WWTPs did not cause significant increase in anti-estrogenic activity in the receiving streams.”
Line 353: “hat” did you mean “had”?

Lin 434: order of words “ …the effluents still had …”

Lines 442-446: Sentence two long – the part that explains unspecific/specific toxicity is a separate idea.

Line 512: wording: …compounds with aryl-hydrocarbon receptor agonistic activity, resulting in…”

Line 524: wording: …where the activity increased by 12.6% which was not significantly different.” (simplify it)

Line 629: wording: …record potential pollution reduction in the receiving waters.”

Line 631: wording: “The findings from our study can be extrapolated…”

Line 632: wording: “The current study demonstrates that the effluents originating from the two WWTPs negatively affect key life cycle ….”

Reviewer 3 ·

Basic reporting

Well done

Experimental design

Well done

Validity of the findings

Well done

Additional comments

None

---

## Round 0.3 · accepted · Accept

Dear Authors,

Thank you for addressing the Reviewer's comments. I have assessed the revision and find the changes sufficient. Thus, the manuscript is now ready for publication.